# Retinotopy Inspired Brain Encoding Model and the All-for-One Training Recipe

## Abstract

Brain encoding models aim to predict brain voxel-wise responses to stimuli images, replicating brain signals captured by neuroimaging techniques. There is a large volume of publicly available data, but training a comprehensive brain encoding model is challenging. The main difficulties stem from a) diversity within individual brain, with functional heterogeneous brain regions; b) diversity of brains from different subjects, due to genetic and developmental differences; c) diversity of imaging modalities and processing pipelines. We use this diversity to our advantage by introducing the All-for-One training recipe, which divides the challenging one-big-model problem into multiple small models, with the small models aggregating the knowledge while preserving the distinction between the different functional regions. Agnostic of the training recipe, we use biological knowledge of the brain, specifically retinotopy, to introduce inductive bias to learn a 3D brain-to-image mapping that ensures a) each neuron knows which image regions and semantic levels to gather information, and b) no neurons are left behind in the model.

We pre-trained a brain encoding model using over one million data points from five public datasets spanning three imaging modalities. To the best of our knowledge, this is the most comprehensive brain encoding model to the date. We demonstrate the effectiveness of the pre-trained model as a drop-in replacement for commonly used vision backbone models. Furthermore, we demonstrate the application of the model to brain decoding. Code and the model checkpoint will be made available.

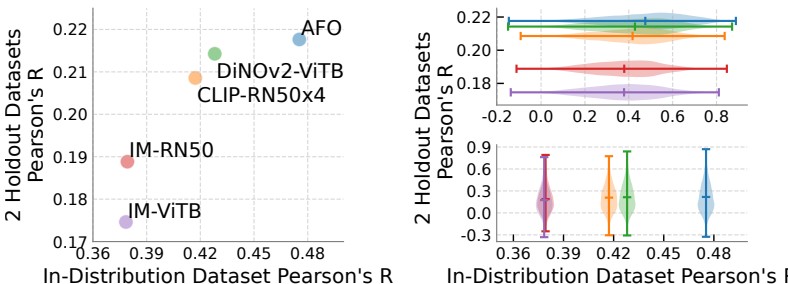

Figure 1: *All-for-One* recipe pre-trained backbone model evaluated by linear probing brain encoding. All models remain frozen, the dimension of latent image features are reduced using PCA to a consistent size. Subsequently, a linear regression is conducted for each voxel. The in-distribution dataset comprises one subject from NSD, the holdout datasets consist of two subjects from BOLD5000 and ThingsfMRI1. Violin plot show distribution of score over voxels.

Submitted to 37th Conference on Neural Information Processing Systems (NeurIPS 2023). Do not distribute.

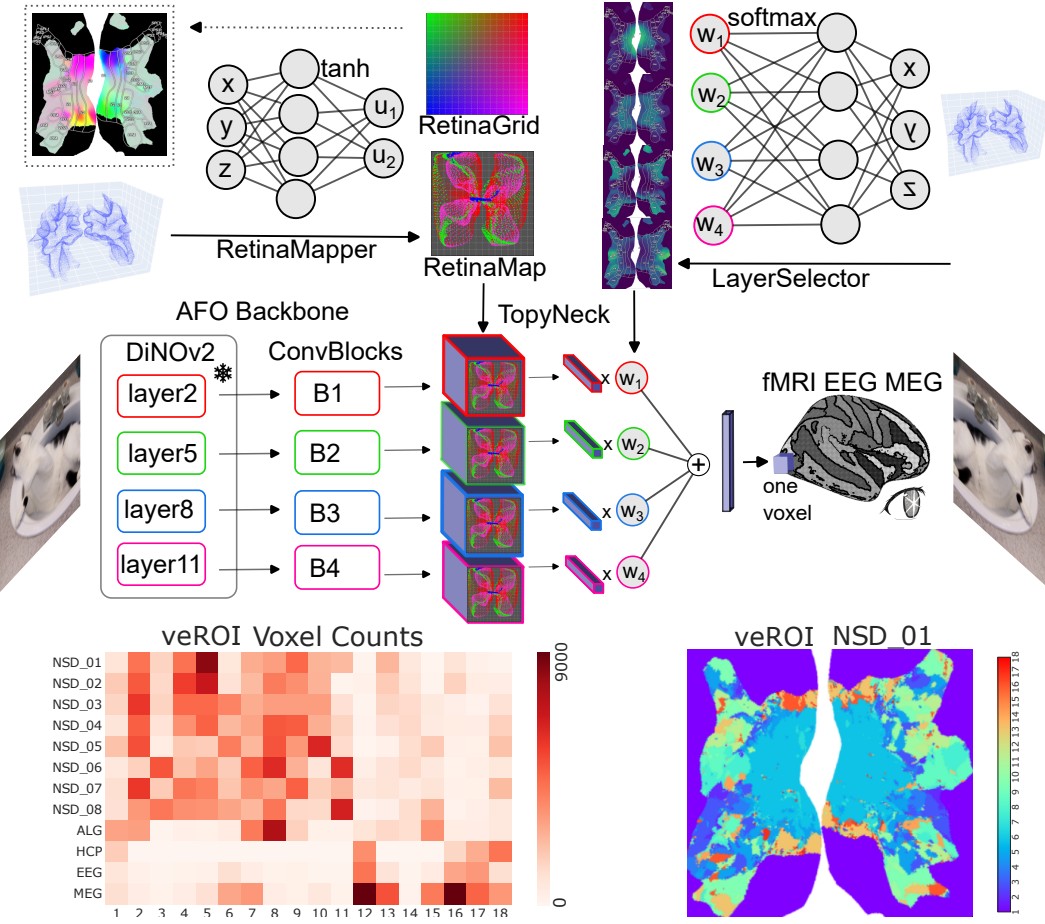

Figure 2: The proposed brain encoding model consists of three main components: the *backbone*, the *TopyNeck*, and the linear regression *head*. The *backbone* is trainable convolution blocks attached to a frozen *DiNOv2-ViT-B* model. *TopyNeck* selects one-dimensional features for each voxel based on its physical coordinates. *TopyNeck* composes of *RetinaMapper* that maps the voxel to a 2D image grid (*RetinaGrid*), and *LayerSelector* that combine feature vectors obtained from backbone layers. Each dot in *RetinaMap* is a voxel, and color corresponds to `argmax` of *LayerSelector*. Finally, a no-weight-sharing linear regression is conducted for each voxel. Voxel-wise encoding ROI (*veROI*), is a novel brain parcellation that unifies multi-modal subjects.

## 1 Introduction

There is a growing body of research in neuroscience that utilizes brain encoding models. The model predicts voxel-wise brain response to visual stimuli, and it can be depicted as a multi-task regression problem where each voxel is a task. The brain encoding model serves as a computational counterpart to biological brains Wen et al. (2018). The common practice for building brain encoding models is to use pre-trained models from image classification Deng et al. (2009), text-to-image alignment Radford et al. (2021), or self-supervised tasks Oquab et al. (2023). These pre-trained models may excel at their benchmarked task; however, Schrimpf et al. (2018) show that the image-classification benchmark score does not align with prediction performance in brain encoding.

Building a model from all data sources poses a significant challenge due to heterogeneity in data: a) diversity in functional sub-modules within each brain, b) genetic and developmental differences across subjects, c) inconsistent imaging techniques and pre-processing pipelines. The current best practice is to build Region-of-Interest (ROI)[1] models over subjects from the same dataset Cichy et al. (2021) Willeke et al. (2022) Allen et al. (2022), where ROIs are predefined by well-studied anatomical and functional properties of the brain voxels. However, the ROI-model approach lacks the potential benefits for ROIs to aggregate knowledge and collaborate. This issue can be mitigated to some extent

---

[1]ROI refers to brain atlas parcellations

by adjusting the granularity of ROIs. This work proposes a multi-stage All-for-One (AFO) training recipe that explicitly lets ROIs aggregate knowledge while keeping the main training objective less challenging than training for one all-ROI model. Borrowing the idea of 'Dark knowledge' distillation Hinton et al. (2015), we use denoising to ensure the aggregated knowledge is clean.

Biological domain knowledge of the brain, specifically retinotopy, can be explored to design a better model Lurz et al. (2021). The retina cells are physically wired through the optic nerve to the lateral geniculate nucleus, which connects to the visual cortex. Thus, visual cortex cells preserve the topological structure of images projected to the retina. This study explicitly defines a *RetinaMapper* function that replicates retinotopic mapping. An obvious solution is learning a forward mapping that transforms 2D *RetinaGrid* into a neuron in a 3D brain location. However, such forward mapping can not guarantee to be surjective: every 3D neuron location is the mapped from at least one 2D *RetinaGrid*. Our solution is to model the *RetinaMapper* from the inverse perspective, mapping 3D neuron to 2D *RetinaGrid*. *RetinaMapper* is learned without ground-truth supervision, but still exhibits retinotopic behavior, as shown in our results.

A well-reported phenomenon is that neuron voxels are mapped to shallow to deep layers of a feed-forward neuron network Takagi and Nishimoto (2022). This motivates the common practice of selecting the best layers for each voxel. But per-voxel hyper-parameter tuning is highly noisy and prone to overfitting; previous studies overcome this by choosing the same layers for each ROI. In this study, we propose a *LayerSelector* module that enforces spatial proximity, thus allowing a flexible and robust selection of layers.

## 2 Related work

The field of computational neuroscience has been actively exploring the task of brain encoding, highlighting from Kay et al. (2008) Naselaris et al. (2011), surveyed by Wen et al. (2018). There are several initiatives and benchmarks: The brain-score Schrimpf et al. (2018) initiative compares frozen image backbone models using a PCA and linear regression pipeline. The PCA approach allows for a fair comparison of vision models with different latent dimensions. Additionally, Conwell et al. (2022) utilized a similar frozen PCA pipeline to benchmark various vision models on the NSD dataset. The Algonauts challenge Cichy et al. (2021) benchmarks end-to-end trained model without the constraint of frozen model and PCA dimension reduction. The Sensorium benchmark Willeke et al. (2022) worked on invasive mouse V1 imaging data. The Things initiative Hebart et al. (2023) provides fine-grid image captions which can be used for hypotheses testing. These datasets and benchmarks cover a wide range of imaging modalities, and preprocessing and denoising pipelines Kay et al. (2013) Prince et al. (2022). The All-for-One training recipe aims to leverage all of these diverse data sources to pre-train a comprehensive brain encoding model.

The neuroscience community has extensively applied brain encoding models to unravel the biological mechanisms underlying brain function. St-Yves et al. (2022) employed transfer learning techniques with brain encoding models to investigate the hierarchical organization of the brain. Franke et al. (2022) applied the model to study color coding in mouse neurons. The NeuroGen framework Gu et al. (2022) combined brain encoding models with image generation models, they utilize gradient-based methods to manipulate stimulus images. Bashivan et al. (2019) generated maximally excited images for populations of neurons and presented these images to subjects to validate the conclusions. On the other hand, there are fruitful studies of brain decoding[2] without a brain encoding model Takagi and Nishimoto (2022) Gu et al. (2023) Lu et al. (2023) Gu et al. (2023). Their framework is to take a pre-trained text-conditioned image generation model Ho et al. (2020) Rombach et al. (2022), then train a mapping function that aligns brain patterns to the text-condition embeddings space. However, we argue that decoding without a pre-trained encoding model is less efficient: Firstly, this pipeline is tightly linked to the pre-trained image generation model. Also, this pipeline face challenges in effectively utilizing heterogeneous data from various imaging modalities. We argue that decoding with a frozen encoding model is more efficient as this approach is agnostic to the specific image generation model.

Previous studies also explored incorporating retinotopy into the brain encoding model. Allen et al. (2022) fits Gabor filters of various sizes and locations for each voxel. Lurz et al. (2021) also employed

---

[2]We use the term *encoding* for mapping from stimuli image to brain voxels, *decoding* for the reverse.

the *RetinaMapper*, but their work focuses on training with the same imaging modality and one single ROI. In contrast, our approach tries to model the whole visual brain with diverse data sources.

# 3 Method

The voxel-wise encoding model (Fig 2) comprises three main components: Firstly, the **backbone** processes the input image and extracts latent image features from its intermediate layers. Next, the **neck** component compresses the feature vector for each voxel. Finally, the **head** applies a linear regression model to fit a prediction for each voxel. Let $M^l \in \mathcal{R}^{D \times \frac{H}{k} \times \frac{W}{k}}$ be the feature map output from the frozen backbone, where $l$ is the layer index, $k$ is the down-scale factor, we refer the $\frac{H}{k} \times \frac{W}{k}$ grid as *RetinaGrid*. The brain encoding model can be formulated as learning a mapping function $\mathcal{F}$ (Eq 1), where $\mathcal{N}$ depends on the imaging modality[3]. $\mathcal{N}_{MRI} := (X \times Y \times Z) \times 1$, $\mathcal{N}_{EEG} := C \times T$, $\mathcal{N}_{MEG} := (X \times Y \times Z) \times T$

$$\mathcal{F} : \mathcal{R}^{(L \times D) \times \frac{H}{k} \times \frac{W}{k}} \to \mathcal{R}^{\mathcal{N}} \tag{1}$$

## 3.1 TopyNeck

**RetinaMapper** The biological retinotopy process is mapping $f : \mathcal{R}^{\frac{H}{k} \times \frac{W}{k}} \to \mathcal{R}^{X \times Y \times Z}$. *RitinaMapper* aims to replicate this mapping. However, $f$ can not guarantee to be surjective: every 3D neuron location is the mapped from at least one 2D *RetinaGrid*. Instead of the forward mapping $f$, we learn a reverse injective mapping $f' : \mathcal{R}^{X \times Y \times Z} \to \mathcal{R}^{\frac{H}{k} \times \frac{W}{k}}$ and use tanh activation function to guarantee the output 2D coordinates lies within the *RetinaGrid*. The *RetinaMapper* is formulated as

$$u = \text{tanh}(\text{MLP}(\text{PE}(p))) \tag{2}$$

where $p \in \mathcal{R}^{\mathcal{N} \times 3}$ is the voxel's spatial coordinate, PE is sinusoidal positional encoding function, $u \in \mathcal{R}^{\mathcal{N} \times 2}$ is coordinates in the *RetinaGrid*. During training, a small non-trainable variance $\sigma$ is introduced $u' \sim \mathcal{N}(u, \sigma)$. At inference time $\sigma$ is set to 0. At each $u'$, linear interpolation is performed to obtain a 1-D feature vector $m^l \in \mathcal{R}^{\mathcal{N} \times D}$ for each layer $l$. Furthermore, Another 1-D feature vector $q^l = \text{MLP}(\text{GlobalAvgPool}(M^l), \text{GlobalMaxPool}(M^l))$ is added to $m^l$. Parameters of *RetinaMapper* is shared for all layers. Figure 2 and 4 show examples of such mapping. The color dots in RetinaGrid indicate which 3D neuron layers it is from. The blank area indicates image regions that are unused for prediction.

**LayerSelector** Early visual to downstream regions have growing receptive field sizes and neurons' latent representation of the stimuli image grows abstract. This motivates matching voxels to layers in feed-forward neuron networks. But selecting the best or top layers for each voxel is suspected to be overfitting. *LayerSelector* enforce spatial proximity formulated as

$$\eta = \text{softmax}(\text{MLP}(\text{PE}(p))) \tag{3}$$

where $\eta \in \mathcal{R}^{\mathcal{N} \times L}$. The 1-D feature vectors sampled from various layers at *RetinaGrid* is reduced as $m_i^* = \sum_L \eta_i^l m_i^l$. Regularization loss $l_{ent} = \sum_L \eta^l \log \eta^l$ is applied to prevent converging to a local minimum that only selects one single layer.

## 3.2 All-for-One training recipe

Dividing neuron voxels into ROIs loses ROIs' potential to aggregate knowledge and collaborate. Mixing can also negatively affect individual voxel performance, making learning more challenging. The AFO recipe aims to gather the benefits from both dividing and mixing. Multiple stages models are trained (Figure 3): In stage one, each ROI model is trained separately. In stage two, each ROI model is trained to distill the dark knowledge Hinton et al. (2015) from all other ROIs, but the ground truth loss is only applied on the target ROI, other ROIs are helpers, and their parameters were discarded after training. Model checkpointing and early stopping are conditioned only on the target ROI. In stage three, the final model is trained with all ROIs as outputs, with dark knowledge and ground truth loss. The final product is one comprehensive all-ROI model.

---

[3]We use a unified term *voxel* to refer to a single smallest element in $\mathcal{N}$.

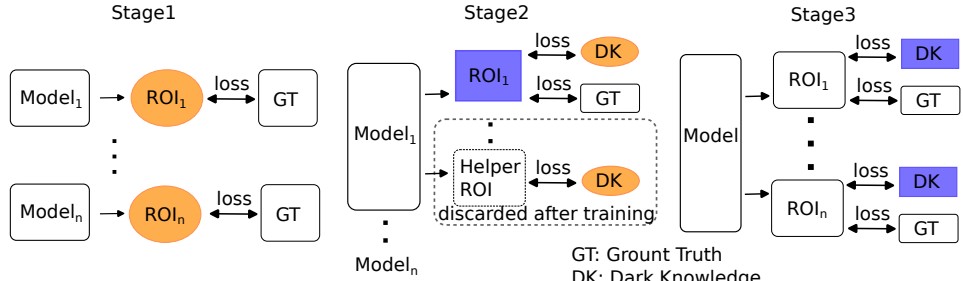

Figure 3: *All-for-One* training recipe involves training multiple stage of models using dark knowledge distillation. In **Stage1**, a separate model is trained for each ROI. In **Stage2**, each model is an all-ROI model that leverages the dark knowledge from all other models as helpers, the parameters of these helper models are discarded after training. In **Stage3**, a single all-ROI model is trained.

Table 1: Brain encoding datasets. The term *Datapoints* refers to the number of image stimulus presentations, including repeated presentation of the same image.

|  | Training Datasets | | | | | Holdout Datasets | |
|---|---|---|---|---|---|---|---|
|  | NSD | HCP MOVIE | Algonauts 2021 | Things MEG1 | Things EEG2 | BOLD 5000 | Things fMRI1 |
| **Datapoints** | 240K | 441K | 30K | 88K | 640K | 20K | 24K |
| **Subjects** | 8 | 184 | 10 | 4 | 10 | 4 | 3 |
| **Voxels** | 315K | 29K | 13K | 60K | 17K | 9K | 19K |
| **Modality** | 7T fMRI | 7T fMRI | 3T fMRI | MEG | EEG | 3T fMRI | 3T fMRI |

## 3.3 Voxel-wise encoding ROI

We need a unified ROI parcellation that is defined for all subjects from various imaging modalities. To generate such a unified ROI, we utilize the final linear regression weight, which is extracted from an average of 10 all-ROI models. We start by performing Euclidean distance k-means clustering on the weights to reduce the dimension of voxel counts. Subsequently, Ward's method applies hierarchical clustering to find the cluster centroids. This hierarchical clustering results in a dendrogram. We cut the dendrogram at a hand-picked threshold to identify the *veROIs*. By adjusting this threshold, we can control the granularity of the *veROIs*.

## 4 Experiments

### 4.1 Datasets

We utilize 7 publicly available datasets for our experiments (Table 1). Details are provided in Allen et al. (2022) Van Essen et al. (2012) Cichy et al. (2021) Hebart et al. (2023) Gifford et al. (2022) Chang et al. (2019). We use only voxels from the visual brain. Each dataset was divided into training, validation, and test sets with a ratio around $90 : 6 : 4$. For the Things datasets, we use repeatedly represented images as the test set. All the experiment results are reported from the test set unless specified. The HCP video was split into chunks of 20 seconds to ensure no data leak, and a time delay of 4 seconds between video frames and fMRI frames was applied Khosla et al. (2021), blank resting-state segments are not discarded. For video stimulus, we extracted frames at a rate of one frame per second. We only use one frame for the ALG dataset.

Notably, except for the NSD dataset, all subjects from other datasets viewed the same set of images. As a compromise for computation intensity, we concatenated the voxels from ALG EEG MEG subjects into each single large brain, voxel's spatial coordinates are placed in an evenly spaced grid. For the HCP dataset, a group average was performed due to the large number of subjects and the lower SNR in each individual subject. All datasets have spatial coordinates for voxels except the EEG dataset, EEG voxel's spatial coordinates are generated from dummy sequential numbers.

### 4.2 TopyNeck probing

**RetinaMapper** In Figure 4, for NSD subjects, early visual voxels were mapped to span most of the *RetinaGrid*, while downstream-region voxels remained concentrated in the center. The ablation study presented in Table 2 further demonstrates the outstanding importance of the *RetinaMapper* for early visual voxels in NSD subjects. This alignment with retinotopy design motivation. However, for other low SNR datasets, no clear retinotopic mapping was observed, suggesting that the *RetinaMapper* may not be necessary in such cases, and a constant mapping to the center could be sufficient.

**LayerSelector** In Figure 5, for subject NSD_01, a smooth transition from shallow to deep layers was observed. This alignment with the design motivation. Ablation study in Table 2 also indicates significant improvement for NSD subjects compared to un-weighted averaging layers or selecting a single layer. However, for low SNR datasets, the trend was to select only the last layer (Figure 4), suggesting that the *LayerSelector* module may not be necessary in such cases.

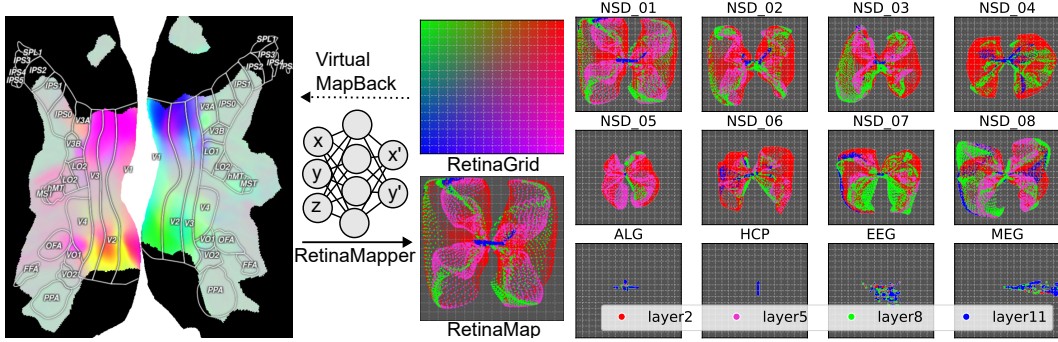

Figure 4: *RetinaMapper* maps voxels to *RetinaGrid*. Each dot on *RetinaMap* is a voxel colored by `argmax` of the *LayerSelector*, colors indicate selection of layers.

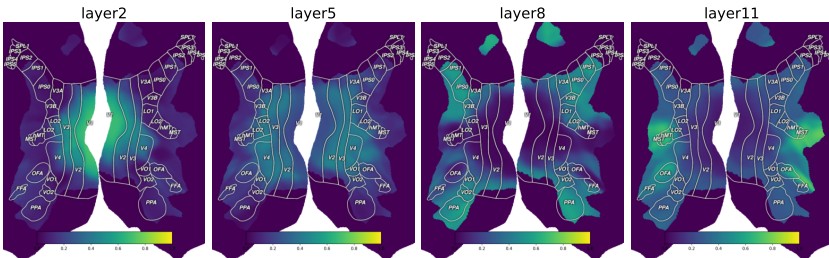

Figure 5: *LayerSelector* re-weights backbone layers, outputs for all layers sum to 1. Results are showed for subject NSD_01.

### 4.3 All-for-One recipe results

In Table 3, a significant performance gap between the S1 and S2 models indicates the effectiveness of aggregating knowledge among ROIs. We also study a randROI that has the exact same number of ROIs and number of voxels inside each ROI. S1 and S2 gap is not observed in the randROI approach, as randROI already covers all types of voxels in every ROI. Furthermore, the model trained with ground truth (NoDK) as helpers shows little to no improvement over the S1 model. This suggests that the quality of the helper ROI is critical for the AFO recipe, as involving noisy helpers makes the training process unnecessarily challenging. In this context, dark knowledge plays a crucial role as denoising. However, solely dark knowledge distillation doesn't have a great impact as can be inferred from the small gap between randROI S1 and S2 models.

Table 2: *TopyNeck* ablation study. The reported numbers are the average Pearson correlation coefficient across all voxels. Results are averaged over three runs. *FrozenRM* maps every voxel to the center, *FrozenLS* outputs uniform weight for each layer. *NoRegLS* selects a single layer.

| Subject | NSD_01 | | | | NSD_08 | | | | EEG |
|---|---|---|---|---|---|---|---|---|---|
| **ROI** | all | early | late | mid | all | early | late | mid | all |
| **FullTopyNeck** | **0.462** | **0.515** | **0.435** | **0.470** | **0.291** | **0.304** | 0.285 | 0.292 | 0.228 |
| FrozenRM | 0.441 | 0.476 | 0.422 | 0.452 | 0.274 | 0.261 | 0.280 | 0.272 | 0.226 |
| w/o GlobalPool | 0.457 | 0.513 | 0.428 | 0.467 | 0.293 | 0.303 | **0.289** | **0.295** | **0.230** |
| FrozenLS | 0.451 | 0.512 | 0.419 | 0.466 | 0.280 | 0.300 | 0.270 | 0.279 | 0.224 |
| NoRegLS | 0.447 | 0.505 | 0.417 | 0.464 | 0.287 | 0.299 | 0.282 | 0.284 | 0.229 |

Table 3: *All-for-One* training recipe ablation study. The reported numbers are the average Pearson correlation coefficient across all voxels, NSD(NC) is the median of noise-normalized score. *NaiveMix* train one all-ROI model. *NoDK* use ground truth as helpers. *randROI* and *veROI* has the exact same size. *S2+1* indicates one extra iteration of stage2. $b$ is number of parameters in the convolution blocks, $n$ is number of voxels, $d$ is feature dimension, $r$ is number of ROIs.

| Method | # Params | Dataset(s) | | | | | | |
|---|---|---|---|---|---|---|---|---|
| | | NSD | EEG | MEG | HCP | ALG | **ALL** | NSD (NC) |
| NaiveMix | $b + nd$ | 0.422 | 0.212 | 0.180 | 0.340 | 0.256 | 0.367 | 0.560 |
| veROIS1 | $rb + nd$ | 0.425 | 0.212 | 0.194 | 0.346 | 0.265 | 0.371 | 0.567 |
| veROIS2 | $rb + nd$ | 0.433 | 0.222 | 0.209 | 0.365 | 0.266 | 0.380 | 0.588 |
| **veROIS3** | $b + nd$ | **0.435** | 0.225 | 0.210 | **0.366** | **0.267** | **0.382** | **0.593** |
| veROIS2+1 | $rb + nd$ | 0.432 | **0.226** | **0.211** | 0.362 | 0.264 | 0.380 | 0.586 |
| NoDK | $rb + nd$ | 0.426 | 0.216 | 0.186 | 0.349 | 0.256 | 0.371 | 0.569 |
| randROIS1 | $rb + nd$ | 0.431 | 0.216 | 0.207 | 0.343 | 0.258 | 0.377 | 0.584 |
| randROIS2 | $rb + nd$ | 0.432 | 0.220 | 0.207 | 0.348 | 0.259 | 0.378 | 0.586 |

## 4.4 veROI results

Figure 6 shows veROI on cortex across all NSD subjects, early visual areas is centered around veROI_5 (blue) and downstream areas centered around veROI_9 (green), voxels that drop out from the field of view in early visual areas are centered around veROI_16 (red). The score for each veROI for subject NSD_01 can be found in Figure 8, where veROI_12 onward is mainly for the low SNR voxels. From the heatmap in Figure 2 we can also observe that veROI_12 onward is mainly HCP, EEG, and MEG subjects.

## 4.5 Brain decoding

**Methods** In this study, brain decoding refers to the task of ranking and retrieving candidate images from a candidate set, retrieved images are to match a given brain response pattern. The decoding pipeline involves forwarding each candidate image through the brain encoding model and measuring Pearson's correlation coefficient between the model's prediction and the ground truth.

**Results** The experiments are conducted on 500 validation images as candidate images. As a qualitative analysis, Figure 7 and Figure 9 demonstrate that when conditioning on the early visual area or veROI_5, texture and orientation are more preserved in the decoded images. Conversely, when conditioning on downstream ROIs, semantic concepts are more preserved. Additionally, Figure 8 shows that image retrieval achieves high accuracy when conditioned on early ROIs. Quantitative exploration of the functional roles of ROIs is beyond the scope of this study. Future work may involve

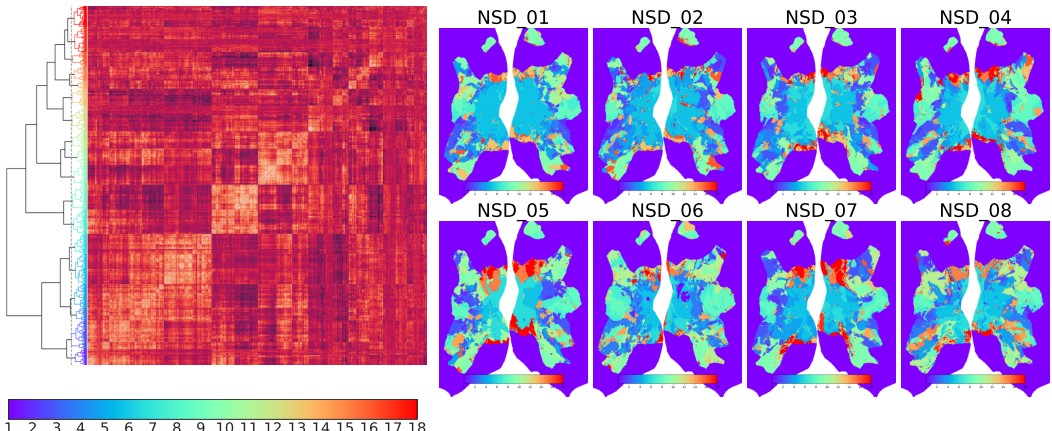

Figure 6: *veROI* cluster voxels into ROIs by hierarchical clustering. ROIs are identified by cutting the linkage at a manually selected threshold value(dashed line). The feature used for clustering is the linear regression weight associated with each voxel.

investigating semantic concepts with image generation models. Furthermore, the gradient of the encoding model can be utilized to facilitate image generation and manipulation.

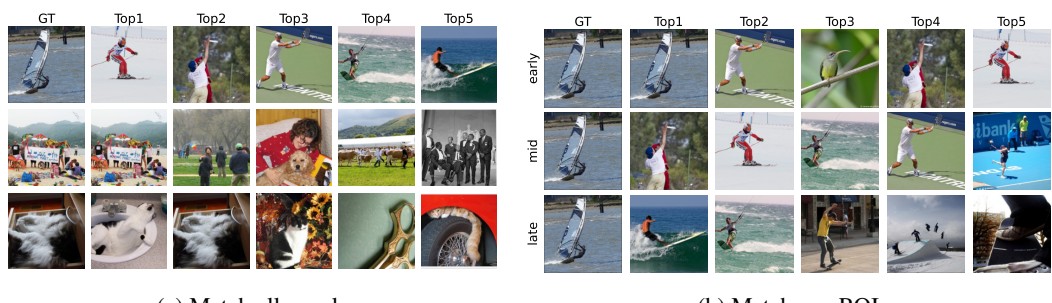

(a) Match all voxels             (b) Match one ROI

Figure 7: Image retrieval to match brain response pattern. Images are ranked by Pearson's r of captured biological brain pattern and model output. Results are for subject NSD_01.

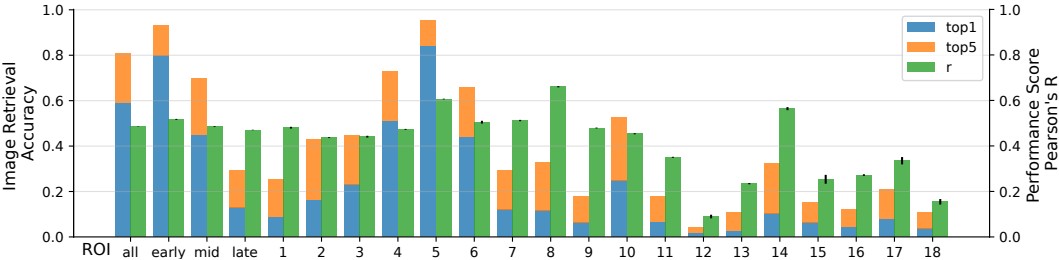

Figure 8: Performance of image retrieval(blue and orange) conditioned on ROIs. The integer numbers are the indices of the *veROIs*. Performance scores of brain encoding(green) are the average value of the voxels within each ROI, standard error is in black. Results are for subject NSD_01.

### 4.6 Implementation details

We use smooth L1 loss with $\beta = 0.01$, regulirazation loss $l_{ent}$ is scaled down by $\lambda = 0.00003$. AdaBelief optimizer Zhuang et al. (2020) is employed with `lr` $= 0.003$, `batchsize` $= 128$, `weight_decay` $= 0.0001$, $(\beta_1, \beta_2) = (0.9, 0.999)$. Notably, we mix subjects in one mini-batch, and the effective batch size for each subject is less than the total. Due to memory constrain, we

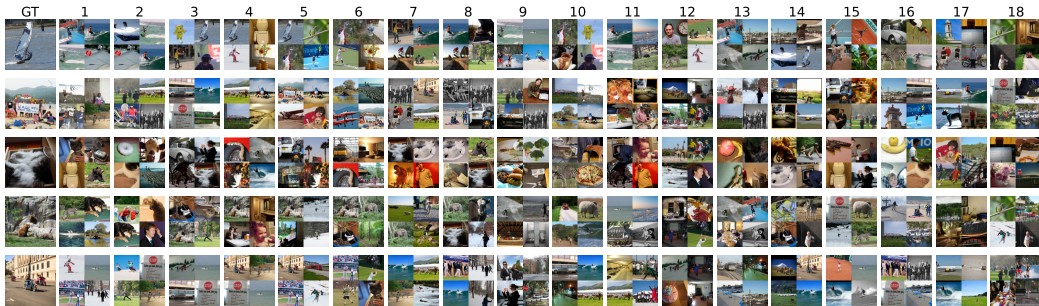

Figure 9: Image retrieval conditioned on *veROIs*. The numerical numbers are the indices of *veROIs*. The top four images are placed from the top left to the bottom right.

randomly sample up to 8000 voxels for each training datapoint, there is 436,715 voxels totaling all subjects. Early stopping is configured with `patience = 20` epochs, we define one epoch as 10% of the total training data. Greedy *Model Soup* Wortsman et al. (2022) is applied at the top 10 validation checkpoints. Backbone is kept frozen except `LayerNorm` running statistics is updated. Input resolution is $224 \times 224$ and the feature from backbone layers are all of the size $768 \times 16 \times 16$. The attached trainable convolution block is three zero-padded 5x5 convolutions with skip connection and `LayerNorm`, $C = 768$. The last convolution layer reduces the dimension to $D = 256$. We trained all models on single NVIDIA RTX 2080 Ti 12GB GPUs at a reduced clock speed of 1140Mhz, single-subject all-ROI models consume half to 1 GPU hour, all-subject single-ROI models consume 3 to 5 GPU hours, all-subject all-ROI models consume 10 GPU hours. The complete AFO recipe total around 300 GPU hours. Models are trained Pytorch Lightning Falcon (2019) mixed precision FP16. To boost training speed, MLPs in *RetinaMapper* and *LayerSelector* are pre-optimized by a single-subject all-ROI model, they are loaded and kept frozen in the AFO recipe, this gives 2 times faster convergence speed.

## 5  Conclusion and Limitations

We proposed the *AFO* recipe alongside *veROI* to address the issue of heterogeneity in publicly available datasets. To the best of our knowledge, our pre-trained model constructed with over 1 million data points is the most comprehensive brain encoding model to date. The model shows superior performance when transferred to small hold-out datasets. As demonstrated by our brain decoding experiments, the pre-trained model could facilitate further neuroscience research.

We also designed *TopyNeck* inspired by retinotopy, which showed retinotopic behavior despite having no ground truth supervision for the retinotopic mapping function. However, the retinotopic behavior diminishes when the target dataset SNR is low, e.g. EEG, MEG. This suggests a simple alternative approach is sufficient in such a case.

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

319     [cs, stat].

