# OpenReview forum: "Retinotopy Inspired Brain Encoding Model and the All-for-One Training Recipe"
_NeurIPS.cc/2023/Conference — Submitted to NeurIPS 2023_

### Official Review · Reviewer_HdGa · 2023-07-02

**Soundness:** 3 good
**Presentation:** 3 good
**Contribution:** 2 fair
**Rating:** 5
**Confidence:** 3

**Summary:**

The authors make two claimed contributions:
1. They propose an architecture for an encoding model. This architecture consists of three key components.
* A frozen DINO trained ViT-B backbone + learnable convolutional layers on top of ViT feature maps
* A differentiable spatial sampling layer (implemented via pytorch grid_sample, similar to [1]), where the 2D coordinates are predicted from 3D voxel coordinates
* A differentiable softmax based layer selector
2. They propose an "All-for-One" training recipe. Which incorporates the following:
* "dark knowledge distillation" (typically referred to as knowledge distillation or network distillation in most other machine learning works), where they use ROI specific networks to train larger networks
* They propose a new parcellation across brains which they call "veROIs", which is extracted via k-means clustering of voxel weights.

To validate this method, the authors visualize the learned spatial sampling grids and how the preferred layer varies across voxels. The authors further perform an ablation study of the encoder, and perform image retrieval using their network.

The authors provide an illustration of how the sampling grid evolves during training in the supplementary.

[1] Jaderberg, Max, Karen Simonyan, and Andrew Zisserman. "Spatial transformer networks." Advances in neural information processing systems 28 (2015).

**Strengths:**

The paper is novel in the combination of techniques, which uses online-learned sampling grids with soft backbone layer assignment to predict the brain activations, this design is reasonably biologically motivated.

The authors proposal of clustering based ROI assignment is also novel in the context of decoding (image retrieval).

The paper provides definitions and dimensions for most variables where appropriate, and the figures are illustrative.

My view is that the proposed decoder architecture is moderately novel in the context of encoding models. There is insufficient information (lack of detail) to judge the all-for-one training.

**Weaknesses:**

The paper is interesting, however my key concern lies in the lack of details for the all-for-one training scheme, and the lack of evaluation in comparing against simple linear-regression + backbones (ResNet, CLIP ViT [1])
* On the clarity of the paper
  * The high level clarity of the paper regarding the all-for-one training is poor and could use significant improvement.
  * It is not clear how exactly the veROIs are used. Is the encoding model ultimately at the voxel-level? If so, are the veROIs used for the network distillation and image retrieval steps? Do you share the backbone and just use independent linear weights for the voxels?
  * It is not clear which subjects you use for Stage 1/2/3. Do you train with one subject's ROI? Or do you train on all subjects' single ROI (using stage 1 as an example). If so, do you take veROI-1 for example, then train a model on all subjects from all modalities for veROI-1?
  * It is not clear how you get the all-ROI model to derive the veROIs. Is this model trained on all subjects and all modalities? Do you train just the last linear weights? Or you use the DINO features? Do you train the conv weights?
  * It is not clear how you train across subjects and datasets where the number of voxels are not the same. This is a central claim in your paper.

* On motivation
   * Currently the justification for repeated distillation is weak. Your experiments show that distillation does help from a performance standpoint, however your motivation differs from the typical use of distillation (which is to accelerate inference using a student model). Could you better motivate this?

* On prior work
  * For the topic of brain encoding models that utilize differentiable spatial sampling, I recommend citing [2,3] which are similarly brain motivated, as well as [4] which is one of the more significant papers that uses differentiable sampling.

* On the soundness of the baseline
  * For the "FrozenRM" encoder, you mention that every voxel is mapped to the center. Since you use DINO based on ViT, there is an extra classifier token (usually the first token). Typically when using ViT based architectures for spatially invariant tasks, this is the embedding you use. I recommend modifying the FrozenRM baseline, or adding an additional baseline when you use this token
  * The same criticism applies to the GlobalPool token, I recommend adding/replacing a baseline where the global pooled representation is replaced with the classifier token

* On the lack of baselines
  * Currently the authors perform ablation studies, but do not perform qualitative or quantitive comparisons against other works [1]. The paper would be strengthened by adding comparisons to linear-regression based single subject voxel-wise encoding models based on different architectures (global pooled ResNet features , VGG, Gabor features, GIST features) trained on ImageNet, or different objectives (CLIP/OpenCLIP/EVA-CLIP, DINO, any of the SSL/Masking work) adapted to a single subject. I don't expect their multi-subject network to necessarily perform better (nor would it be a negative if they perform worse), but some evaluations are still necessary.

* Minor
  * Figure 4 was quite confusing to me. In the retinagrid case, the colors indicate spatial extent. However in the retinamap, the colors indicate layer selection. I would ask that you provide additional clarity here.
  * The notation in Table 3 is quite confusing, you do not specify what veROIsX is. It is implied that it corresponds to what stage of network distillation you use. Please clarify this.
  * Line 199 "regulirazation", minor misspelling.

On balance, the paper is interesting from an architectural standpoint. I would be happy to take another look at the paper if the authors can clarify their training scheme and add additional baselines.

[1]  Conwell, Colin, et al. "What can 5.17 billion regression fits tell us about artificial models of the human visual system?." SVRHM 2021 Workshop@ NeurIPS. 2021.

[2] Mahner, Florian, et al. "Learning Cortical Magnification with Brain-Optimized Convolutional Neural Networks." Conference on Cognitive Computational Neuroscience. 2022.

[3] Jun, Na Young, Greg Field, and John Pearson. "Efficient coding, channel capacity, and the emergence of retinal mosaics." Advances in neural information processing systems 35 (2022): 32311-32324.

[4] Jaderberg, Max, Karen Simonyan, and Andrew Zisserman. "Spatial transformer networks." Advances in neural information processing systems 28 (2015).

**Questions:**

See weaknesses section for questions

**Limitations:**

The authors adequately address the limitations of their model.

---

> ### Author Rebuttal · Authors · 2023-08-10
>
> - RE: On clarity
>
> “details on all-for-one training”: see G2.
>
> “do you share weights”: see G4. We share the backbone for voxels (in each veROI model), while the last linear regression is unique for each voxel.  In addition, each voxel’s input feature $\mathbf{h}_i$ is affected by RetinaMapper and LayerSelector.
>
> “do you train on all subjects”: see G3 pseudocode. We train each veROI model with all subjects from all datasets.  However,  there is a partial partition of veROIs and subjects (fig2 heatmap): NSD and ALG subjects are mainly in the first eleven veROIs (NSD and ALG use GLMsingle [3] data preprocessing), EEG, MEG, and HCP subjects are primarily in the last seven veROIs.   Still, there’s a good amount of overlapping voxels from MEG subjects in veROI-6 and 7, with early visual regions of NSD subjects(fig6, blue).
>
> “derive the veROIs”: we will change “weights extracted from an average of 10 all-ROI models” to “all-subjects all-ROI models”. We train the model with a shared backbone; we do not use a PCA linear regression [1] on the class token. Each voxel is unique in 1) RetinaMapper and LayerSelector 2) the last linear weight. We use only the last linear weight to cluster.
>
> “how do you train when the number of voxels is not the same”: see G4. Our model can predict any subset of voxels, one subject at a time.  We random sample 8,000 voxels per subject at each batch, as described in section 4.6.
>
> - RE: On motivation
>
> See G2.  Naive mixing of all subjects and ROIs produces inconsistent results: it improves some prediction scores while decreasing for others.
>
> - RE: On prior work
>
> '[2]' and '[3]' is added to the Introduction section. The updated method equations (see G4) explicitly now state we use differentiable sampling.
>
> - RE: On the soundness of the baseline
>
> “class token baseline”. We compared five methods: 1) with GlobalPool, 2) with ClassToken, 3) with both ClassToken and GlobalPool, 4) only GlobalPool, and 5) only ClassToken.  We kept the ConvBlock to the 768x1x1 class token. In 4) and 5), the layer selector is replaced with averaging four layers. ClassToken performs significantly better than GlobalPool (0.456 vs. 0.429), especially on early visual (0.506 vs. 0.394). The best model is with both ClassToken and GlobalPool. We will include the updated model in the code and model weights release. Thank you for pointing this out.
>
> | subject | NSD_01 |  |  |  | NSD_08 |  |  |  | EEG |
> | --- | --- | --- | --- | --- | --- | --- | --- | --- | --- |
> | roi | all | early | late | mid | all | early | late | mid | all |
> | FullTopyNeck(GlobalPool+ClassToken) | **0.467** | 0.508 | **0.448** | **0.471** | **0.294** | **0.308** | 0.286 | **0.299** | 0.224|
> | FullTopyNeck(GlobalPool) | 0.462 | **0.515** | 0.435 | 0.470 | 0.291 | 0.304 | 0.285 | 0.292 | **0.228** |
> | FullTopyNeck(ClassToken) | **0.467** | 0.509 | 0.447 | **0.471** | **0.294** | 0.305 | **0.288** | 0.298 | 0.225 |
> | only GlobalPool | 0.429 | 0.394 | 0.444 | 0.433 | 0.254 | 0.228 | 0.268 | 0.251 | 0.198 |
> | only ClassToken | 0.456 | 0.506 | 0.431 | 0.465 | 0.290 | 0.303 | 0.285 | 0.291 | 0.227 |
>
> - RE: On the lack of baselines
> “benchmark against other backbones”:  we will add a sentence in the Results section to reference Figure 1, which is the benchmark you mentioned. We will add, “All-for-One recipe trained backbone showed superior performance on held-out datasets, benchmarked with PCA linear probe [1] (Figure 1), compared with existing models of Imagenet, CLIP, and DiNOv2 training objective; Resnet and ViT architectures.”
>
> - RE: Minor
>
> “Figure 4”: we will update Figure 4 caption “RetinaGrid color corresponds to that color in the cortex, RetinaMap color does not match the color in RetinaGrid.”.   In RetinaMap, each dot is a voxel mapped from the 3d brain.  We construct a 2D gradient colormap (RetinaGrid) representing pixel location by anchoring four colors on the corners of the image grid.  Each voxel on RetinaMap then uses its mapped pixel location to obtain its corresponding RGB value and assigned it to the cortex vertex.
>
> “Table 3”: we will update the caption “veROISX is stage X”.
>
> “regularization” misspelling will be fixed. Thank you for pointing this out.
>
> ---
>
> [1] What can 1.8 billion regressions tell us about the pressures shaping high-level visual representation in brains and machines?
>
> [2] A massive 7T fMRI dataset to bridge cognitive neuroscience and artificial intelligence, in Nature Neuroscience.
>
> [3] Improving the accuracy of single-trial fMRI response estimates using GLMsingle

---

> > ### Comment · Reviewer_HdGa · 2023-08-11
> >
> > I have read the response by the authors, and have decided to retain the original score (borderline accept).
> >
> > I applaud the authors for performing the class-token experiments, and I find the results very interesting. I also appreciate the clarifications the authors provided in this response, and in the general response.
> >
> > Generally speaking, while **the paper is broadly interesting, I do not consider the paper well written.**
> >
> > I think there needs to be substantial work by the authors to improve the communication of their ideas, deliver clarity in their captions, and provide additional justification for their work. The PDF cannot be revised at the current stage, and this comment was more directed towards future work by the authors.

---

### Official Review · Reviewer_1xvD · 2023-07-04

**Soundness:** 2 fair
**Presentation:** 2 fair
**Contribution:** 3 good
**Rating:** 4
**Confidence:** 4

**Summary:**

This paper develops an all-for-one training model to address the challenge of one big-model problem by converting it into multiple small models, in which the small models aggregate the knowledge while preserving the distinction between the different functional regions. With the proposed method, biological knowledge of the brain, particularly retinotopy, is used to introduce inductive bias into a 3D brain-to-image mapping that ensures a) each neuron knows which regions and semantic levels to gather information, and b) no neurons are left out. Overall, it is an interesting paper, however, there are several concerns about the machine learning novelty, validating the empirical studies, and the clear presentation of the proposed method.


**Strengths:**

Please refer to the question section


**Weaknesses:**

Please refer to the question section


**Questions:**

The followings are the major concerns and minor comments:


1) It is still unclear to me as to what the novelty of this paper is in terms of machine learning. There may be some contributions to computational neuroscience in this paper; however, which part of the machine learning approach is new in this paper?


2) My second concern is the design of the empirical studies. Even though the study contains several beautiful figures (and a movie in supplementary material), more numerical analyses would be helpful to convince the reader that the proposed method is effective. There is a lack of regular analysis and conventional machine learning metrics (such as accuracy, dice, etc.) which make it difficult to understand the results. Specifically, Pearson’s correlation is not an accuracy metric to evaluate the proposed model.


3) The proposed method should be benchmarked in comparison with related state-of-the-art techniques.


4) In this paper, the notations are confusing. In regular papers, scalers are denoted by small letters, vectors are defined with small letters (highlighted in bold), and matrices are denoted by capital letters using bold. In this paper, there are a lot of conflicts. It is so hard to trace what is a set, a matrix, or even a distribution.


5) The proposed method can be summarized in the form of an algorithm or pseudocode.


6) There are some minor linguistic and typo problems in this paper.



**Limitations:**

Please refer to the question section

---

> ### Author Rebuttal · Authors · 2023-08-10
>
> Q1: ``what's the novelty of this paper in terms of machine learning”,
>
> The brain prediction task could be considered a few-shot dense 3D prediction task.  This task requires generalization with high SNR data.  In building a per-voxel prediction, there are far more parameters than training samples, and the model must learn to generalize across subjects, brain ROIs, and sensing modalities, all under noisy measurements.  Naive mixing of the training signals leads to poor results.
>
>
> 1. the all-for-one method shares the same philosophy with sparse coding: we wish to use an over-complete set of voxels to represent (back to) latent image features. An alternative approach is few-shot learning: 300M parameters for each voxel (10K data). The all-for-one recipe could be applied to other multi-task learning problems; each voxel represents a task.
>
> 2. we are trying to learn 2d image to 3d brain voxel dense correspondence without supervision.  It could be viewed as a weakly supervised task: the biological knowledge of retinotopy tells us that the left brain is mapped to the right field of view and vice versa. However, detailed mapping of image regions to brain voxels could be very complex and is largely unknown.  We trained the RetinaMapper without supervision and showed the learned mapping matches retinotopy.
>
>
> Q2: “Pearson’s correlation is not an accuracy metric”: see G3.  Large-scale benchmarks of brain encoding models use Pearson’s r as a reliable metric because it averaged over a large number of voxels.   We measured the mean standard deviation for our experiment as 0.00238; we updated the tables to include the mean standard deviation.
>
> Q3: “evaluate against state-of-the-art”, see G1 and G3.
>
> Q4: “math notations”, see G4. We resolve a conflicting use of $\mathcal{N}$: N (voxel matrix) changed to capital italic bold, N (normal distribution) to \matchcal, vectors changed to bold italic small letter, scalar is small italic letter.
>
> Q5: “pseudocode”, see G2 and attached PDF for the pseudocode.
>
> Q6: "typo": we fixed the typo in Line 199.

---

> ### Comment · Reviewer_1xvD · 2023-08-20
>
> Hello everyone. I have read all the comments and the corresponding responses. The main idea seems interesting in this paper, but a few concerns are not addressed correctly, even in the rebuttal. I believe that this paper needs an additional revision stage to be ready for the publication process. I keep my score as it is and tends to reject this paper. However, I am open to other opinions as well.

---

> > ### Author Response · Authors · 2023-08-20
> >
> > Hi Reviewer 1xvD. Could you provide more details on the concerns not addressed correctly in the rebuttal? Your feedback is valuable and we try to improve in future revision.

---

### Official Review · Reviewer_n3y9 · 2023-07-05

**Soundness:** 2 fair
**Presentation:** 3 good
**Contribution:** 2 fair
**Rating:** 4
**Confidence:** 2

**Summary:**

The manuscript addresses the challenge of generating brain encoding models (specifically for visual stimuli) - which seeks to predict brain responses at the voxel level to visual stimuli. A challenge facing brain encoding is heterogeneity in data modality, individual variability, and functional differences across brain region. Existing models often address the problem of heterogeneity by fitting separate models for different brain regions. However, the authors seek to to fit a single model that encompasses the entire visual brain by leveraging a dark knowledge distillation method in which each ROI distills the dark knowledge present in the other ROIs.  The authors evaluate their method on a variety of functional imaging datasets spanning fMRI, MEG, and EEG.

**Strengths:**

Originality: The authors developed a novel training pipeline utilizing dark knowledge distillation in order to allow ROIs to collaborate during training. The authors also develop a new method for incorporating retinotopy, a biologically realistic feature, into their encoding model.

Quality: The model seems to work across different data modalities (e.g. EEG, MEG, and fMRI), making it broadly applicable.

Clarity: The manuscript provides several ablation studies that investigate the limitations of the approach and which features of the architecture are important for improvement.

Significance: The study introduces a new approach for creating visual brain encoding models from functional imaging data. Their approach could be employed by other groups studying non-visual stimuli as well.

**Weaknesses:**

Unless I am mistaken, the authors do not evaluate their model against the state of the art in  computational speed, making it hard to evaluate whether their model represents a significant improvement, as the improvement in correlation appears to be modest. Furthermore, the message of the paper is a little unclear - what is the main breakthrough the authors are trying to present: the all-for-one training, retinotopy, or ability to work with multiple data modalities. Evaluation against state of the art should occur for each of these topics. Finally, the model is trained on data across fMRI, MEG, and EEG modalities, but the held out data consists only of fMRI data, making evaluation of model generalizability difficult.

**Questions:**

See weakness section for list of suggestions/questions.

**Limitations:**

Limitations are addressed except for points raised earlier about comparison to state of the art.

---

> ### Author Rebuttal · Authors · 2023-08-10
>
> W1: “compare SOTA with computational speed”: see G1.  We build on top of the state-of-the-art retinotopy module and remove the need for an additional mechanism to infer individual voxel's receptive size. Our LayerSelector significantly reduced the computation cost.
>
> W2: “improvement in correlation appears to be modest”: see G3; we will update tables to include the standard deviation of the score (0.002).
>
> W3: “compare to SOTA of each topic”: see G3 and G1.
>
> W3: “what’s the main breakthrough?”:
>
>
> Our retinotopy module (RetinaMapper and LayerSelector) demonstrated generalization ability even when 1) dataset SNR is high and 2) voxels span diverse receptive field sizes and locations.  In addition, the all-for-one recipe performed well when combing diverse voxels and subjects.  Our model is also explainable.
>
>
> W4: “lack held out EEG dataset make evaluation of model generalizability difficult”:
>
> We can evaluate the model's generalizability even if all training and holdout datasets come from fMRI data. Data collected from each research group differ significantly in 1) scanner model and parameters, 2) experimental design, and 3) pre-processing and denoising. For experimental design: NSD presents each image for 3 sec followed by a 1-second blank, while BOLD5000 presents each image for 1 sec followed by a 9-second blank. For denoising: NSD, ALG, and BOLD5000 use GLMsingle [1], while ThingsfMRI uses their own GLM noise regressor because lack of repetition needed for GLMsingle.
>
> ---
>
> [1] Improving the accuracy of single-trial fMRI response estimates using GLMsingle

---

### Official Review · Reviewer_ARqu · 2023-07-06

**Soundness:** 2 fair
**Presentation:** 2 fair
**Contribution:** 3 good
**Rating:** 6
**Confidence:** 3

**Summary:**

The authors tackle the brain encoding task, which predicts brain voxel-level responses to image stimuli. The authors aim to train a comprehensive brain encoding model using the vast amount of public data from diverse imaging modalities and numerous participants. The proposed method, the All-for-One training recipe, divides the one-big-model to multiple small models and aggregates the knowledge together in inference time. An intriguing technique the authors propose is to use retinotopy to introduce inductive bias to learn the mapping.

**Strengths:**

1. The figure quality (especially aesthetics) is not often seen in NeurIPS submissions. Take a look at Figure 2, 4, 5 and 6. Those figures are on par with Nature family submissions.
2. Decent ablation studies.

**Weaknesses:**

1. The design choice is not the most straightforward to process. I have some trouble understanding the logic behind the three-staged design of the proposed All-for-One recipe — specifically, why is it necessary to have these 3 stages?
2. RetinaGrid and RetinaMap seems a bit far-fetched. By far, my impression is that the authors are simply trying to draw an analogy from image formation process in the retina and provide a fancy visualization. The authors are welcome to defend with explanations.
3. Lack of other alternative baselines.

**Questions:**

1. It is a little bit weird to see an MEG and an EEG dataset in 5 other fMRI datasets. Did you conduct experiments to see if the inclusion of MEG and EEG is significantly harming the performance?
2. What is the rationale for enforcing a strong prior on one-to-one mapping between the brain voxel-level responses and the pixels in the image feature maps?
3. Do you think it will be beneficial to include some brief summaries/teasers in the Abstract and/or Introduction regarding the brain decoding investigations?

**Limitations:**

Nothing that I am aware of.

---

> ### Author Rebuttal · Authors · 2023-08-10
>
> S1: Thank you for the nice compliment on our figures!
>
> W1: “how all-for-one works”: see G2 and attached PDF.
>
> W2: “how RetinaMapper works”: see G1. We will update Figure 4 caption “RetinaGrid color corresponds to that color in the cortex, RetinaMap color does not match the color in RetinaGrid.”.
>
> W3: “lack of baseline”: see G3.  We build on top of state-of-the-art and provide ablation study of RetinaMapper, LayerSelector (table 2), and all-for-one recipe (table 3).
>
> Q1: “mixing fMRI with EEG/MEG”: see G2.  Mixing EEG with NSD decreases the performance of both EEG and NSD, mixing EEG with MEG also decreases performance, but combining all five datasets could increase and decrease performance depending on the subjects.
>
> Q2: “why one-to-one mapping”: see G1.  We use LayerSelector to capture different receptive field sizes, compared to one-to-many (i.e. attention mask), one-to-one RetinaMapper is: 1) has less learnable parameters, and 2) directly interpretable.
>
> Q3: “brain decoding teaser”: we briefly mentioned brain decoding in the abstract. We will add that "decoding reveals diverse functionality on brain regions."

---

> > ### Comment · Reviewer_ARqu · 2023-08-18
> > **Response to Rebuttal**
> >
> > I have read the rebuttal by the authors and found the additional experiments provided. I agree with Reviewer HdGa that the content is broadly interesting while the presentation is suboptimal. I have increased the score to 6.

---

### Official Review · Reviewer_CWUX · 2023-07-27

**Soundness:** 3 good
**Presentation:** 4 excellent
**Contribution:** 2 fair
**Rating:** 6
**Confidence:** 2

**Summary:**

This paper focuses on the task of brain encoding model, which aims to predict brain voxel-wise responses to stimulus images. First, the paper proposes the All-for-One (AFO) training recipe, which enhances interactions among multiple ROI models to handle the large diversity within the data. Second, the paper introduces the RetinaMapper to learn a 3D brain-to-image mapping and the LayerSelector to selectively merge features from multiple layers. Finally, the paper trains the model on a very large scale of data and demonstrates the effectiveness of the model through extensive qualitative and quantitative analysis.

**Strengths:**

1. The overall motivation is clear, and the techniques used are straightforward and reasonable. For example, enforcing cross-model interaction/distillation is a reasonable way to enhance each model, and selectively merging information from multiple layers with different receptive field sizes is sensible.

2. The experiments are extensive, and adequate qualitative and quantitative analysis is provided. For example, Table 2 demonstrates the effectiveness of TopyNeck, and both Figure 4 and Figure 5 show the effectiveness of the LayerSelector.

3. The work pre-trains the model in large-scale data, which is impressive.

**Weaknesses:**

1. need to provide more details and insights into specific techniques
2. missing more explanations/references
3. some writing issues

**Questions:**

1. Although the overall motivation is clear, many details and insights of specific techniques are missing and require more explanations/references. For example:

1） It is unclear why the authors introduce non-trainable variance to 'u' and add global features to 'm^l'.

2） The authors do not explain why selecting the best or top layers for each voxel is suspected to be overfitting and how their method addresses this problem.

3） The regularization loss presented in L119 only increases/decreases entropy, and it is unclear how it can prevent convergence to a local minimum that only selects one single layer. The authors should refer to the IM loss mentioned in [1] for the correct term.

4） The details about the distillation in L267 are missing, and the authors should present the concrete form of the distillation loss.
5） Some hyper-parameters are missing due to the lack of details mentioned above, such as the weights of the regularization loss and distillation loss.

[1] Do We Really Need to Access the Source Data? Source Hypothesis Transfer for Unsupervised Domain Adaptation, in ICML 2020.

2. Some of the presentations need improvement:

1) The abbreviations and notations used are numerous, and it would be helpful to simplify them. For example, RetinaMapper, RetinaGrid, AFO, and TopyNeck. The notations used in the experimental section, such as in Table 2 and Table 3, are also confusing and should be simplified.

2) Figure 2 is unclear, and the authors should clearly label which parts correspond to the backbone, TopyNeck, and head.

**Limitations:**

The authors have addressed the limitations

---

> ### Author Rebuttal · Authors · 2023-08-10
>
>
> Q1A: “why add non-trainable variance to 'u’”: see G4. Non-trainable variance is a reparameterization trick.
>
> Q1B: “why add global features to 'm^l’”: see G4. High-level visual voxels are mapped to the center point on the image grid, but their receptive field should cover the whole image; adding 'm^l’ allows information about the entire image to flow through.
>
> Q2 Q3: "regularization loss": we will update the text in the Methods section from “converging to a local minimum that only selects one single layer.” to “an early convergence to a vanishing gradient point, which selects a singularity layer for all voxels”.  In detail: the use of softmax in LayerSelector leads to a vanishing gradient at the value of [1, 0, 0, 0]; our entropy regularization loss, as well as the IM loss mentioned in '[1]', is minimized at a value of [0.25, 0.25, 0.25, 0.25], this will prevent moving towards vanishing gradient. In practice, w/o regularization, we see LayerSelector converge into the same layer for all voxels during the first few epochs and keep stuck for the rest of the training; this leads to a significant performance drop (0.462 vs. 0.447, table 2).
>
> Q4: “distillation loss term”: see G2.
>
> Q5: “hyper-parameters”： regularization loss weight is 0.00003 as described in sec 4.6; distillation loss is decaying, the max_step for decay is 120, we added it to sec 4.6;
>
> “presentations”: we will update the Table 2 and Table 3 captions and the dashed brackets in Figure 2 to show the backbone, neck, and head.

---

> > ### Comment · Reviewer_CWUX · 2023-08-21
> > **Response from Reviewer CWUX**
> >
> > I would like to express my gratitude for the diligent efforts made by the authors in addressing my questions. The authors have addressed my concerns, and I will increase my score to weak accept.

---

### Author Rebuttal · Authors · 2023-08-10

# General response
## G1.
RetinaMapper is a crucial innovation of our method. It is an instance of a retinotopy module in the following discussion.

### A. state-of-the-art retinotopy module (RetinaMapper):
1)  Attention mask per voxel, GNet in [1], requires way more learning parameters than our RetinaMapper, making it: 1) computationally expensive, 2) prone to overfitting. Furthermore, our LayerSelector can infer receptive field sizes from the network layer without the costly estimation of the size of the attention mask.

2) The work of [7] developed a similar mapping as our RetinaMapper. However, we found several issues in [7]:

    1. [7]' activation function is not constrained to the image domain of [-1, 1].
    2. [7] used learnable sigma, which converges to a tiny value (sigma < 1e-6). We fix sigma=0.01.
    3. [7] did not use position encoding.

3) Missing retinotopy module: state-of-the-art in [2] does not use a retinotopy module due to low SNR fMRI data; we also found our RetinaMapper ineffective in that dataset (Figure 4, ALG) because the mapping is trivial: all voxel to the image center. State-of-the-art in [3] also lacked the retinotopy module due to sparse brain location sampling.

### B. computation analysis:

Our RetinaMapper substantially reduces computation costs.  Using per-voxel training with an attention mask (Gabor filter or GNet [1]), the retinotopy module consumes 7.5x more compute (128.8G FLOPs, Table a6) relative to the backbone ViT-B (17G FLOPs).  With RetinaMapper, we use 0.75x compute (12.5G).

### C. Limitations of SOTA:

1.  Mapping of [7] must include an additional mechanism to capture receptive field size variation across the brain voxels.  We combine it with LayerSelector to achieve this goal with minimal computation overhead.

2. one-to-many mapping, as in attention mask, could be used in our RetinaMapper, but it is computationally intense.

Our one-to-one RetinaMapper combined with LayerSelector requires much fewer learning parameters and is more explainable.


## G2.
Regarding the “all-for-one” recipe, motivation, and implementation details:

### A. Motivation:

We observe that combining data across the dataset could improve prediction scores for some but decrease for others, depending on ROIs and subjects.  Table a1 (attached PDF) shows that combining ROIs decreased the score for early and mid ROIs but increased the score for late ROIs. Tables a2 to a4 show that mixing EEG with MEG, EEG with NSD subjects decreased most subjects' scores, and mixing subjects within NSD increased the score for NSD_01. Table a5 shows that combining subjects across five datasets (NSD, EEG, MEG, ALG, HCP) compared to within one dataset has diverging effects.

Our motivation for AFO is:
1. “stage 1” maximizes the score of voxels that prefer trained alone (e.g., early visual ROI),
2. “stage 2” maximizes the score of voxels that prefer trained with others (e.g., high-level ROI),
3. “stage 3” is to distill a single model.

### B. Implementation details:

Pseudocode of the all-for-one is shown in the attached PDF (Listing 1).

## G3.

### A. metric reliability:
The average standard deviation of metric in G2 is 0.00238, measured by 45 configurations and 3 random seeds each.  While each voxel’s r value can vary by chance, an average across 40k voxels makes it statistically reliable.

Benchmarks in 2018 [3] and 2019 [5] use representational similarity analysis but found it to be less consistent across different test sets. Recent benchmarks 2021 [2] and 2022 [4] use averaged voxel-wise Pearson’s r.

### B. Evaluate against state-of-the-art in each topic:

1. Retinotopy module: we combined RetinaMapper with LayerSelector to substantially reduce the learnable parameters making it less prone to overfit, more computationally efficient, and explainable.
2. Recipe: our all-for-one training includes state-of-the-art ROI model methods [2] [3] [4] [5] in stage 1. It could be extended by using ensemble techniques.


## G4.

We updated the equations to clarify that: 1) the model share backbone weights for all voxels, 2) non-trainable variance sigma is a reparameterization trick, and 3) we use differentiable bilinear interpolation sampling.

$\boldsymbol{M}^j = \texttt{ConvBlock}^j(\texttt{ViT}^j(\boldsymbol{X}))$

${\hat{\boldsymbol{h}}}^j = \texttt{AvgMaxPool}(\boldsymbol{M}^j)$

$\boldsymbol{\epsilon}_i \sim \mathcal{N}(\boldsymbol{0}, \boldsymbol{I}_2)$

${\tilde{\boldsymbol{h}}}_i^j = \texttt{INTP}({\boldsymbol{u}_i} + \sigma\boldsymbol{\epsilon}_i, \boldsymbol{M}^j)$

$\boldsymbol{h}\_i = \sum_{j=1}^{4} {\eta}_i^j ({\tilde{\boldsymbol{h}}}_i^j + {\hat{\boldsymbol{h}}}^j)$

$y_i = \boldsymbol{h}^{\boldsymbol{T}}_i \boldsymbol{\beta}_i + b_i$



$j$ is index of 4 candidate layers, $i$ is index of voxels, $\boldsymbol{M}^j \in \mathcal{R}^{16 \times 16 \times 256}$ is feature matrix, ${\hat{\boldsymbol{h}}}^j \in \mathcal{R}^{256}$, $\boldsymbol{\epsilon}_i \in \mathcal{R}^2$ is reparameterization trick.  ${\tilde{\boldsymbol{h}}}_i^j \in \mathcal{R}^{256}$, $\texttt{INTP}$ is bilinear interpolation. $\boldsymbol{u}_i \in \mathcal{R}^{2}$ is RetinaMapepr output, $\boldsymbol{\eta}_i \in \mathcal{R}^{4}$ is LayerSelector output.

---

[1] A massive 7T fMRI dataset to bridge cognitive neuroscience and artificial intelligence, in Nature Neuroscience.

[2] The Algonauts Project 2021 Challenge: How the Human Brain Makes Sense of a World in Motion. *ArXiv:2104.13714*.

[3] Brain-Score: Which Artificial Neural Network for Object Recognition is most Brain-Like? *bioRxiv 407007*

[4] The Sensorium competition on predicting large-scale mouse primary visual cortex activity. *ArXiv:2206.08666*.

[5] The Algonauts Project: A Platform for Communication between the Sciences of Biological and Artificial Intelligence. *ArXiv:1905.05675*.

[6] Sensorium Competition - NeurIPS 2022 Workshop

[7] Generalization in data-driven models of primary visual cortex, in ICML 2021

---

### Decision · Program_Chairs · 2023-09-21

**Decision:**

Reject

**Comment:**

The paper presents a novel training pipeline utilizing dark knowledge distillation in order to allow ROIs to collaborate during training.
The motivation for this contribution is clear, and the techniques used are straightforward and reasonable.
The model seems to work across different data modalities (e.g. EEG, MEG, and fMRI), making it broadly applicable.
Moreover, selectively merging information from multiple layers with different receptive field sizes is sensible.
The experiments are extensive, and adequate qualitative and quantitative analysis is provided. For example, Table 2 demonstrates the effectiveness of TopyNeck, and both Figure 4 and Figure 5 show the effectiveness of the LayerSelector.
The work pre-trains the model in large-scale data, which is impressive. The ablation study is informative.
The figure quality is excellent.

The paper however shows small issues
* An evaluation against the state of the art regarded computational speed is needed.
* the main message of the paper is a bit unclear
* Experiments are limited, e.g. the held out data consists only of fMRI data
* There is a lack of other alternative baselines.

Overall, this is an interesting papers, whose main limitation lies in the writing quality